# The Antidepressant Activity of a Taurine-Containing Derivative of 4-Phenylpyrrolidone-2 in a Model of Chronic Unpredictable Mild Stress

**DOI:** 10.3390/ijms242316564

**Published:** 2023-11-21

**Authors:** Denis A. Borozdenko, Darya I. Gonchar, Vlada I. Bogorodova, Dmitri V. Tarasenko, Evgeniya P. Kramarova, Svetlana S. Khovanova, Yaroslav V. Golubev, Nina M. Kiseleva, Tatiana A. Shmigol, Aiarpi A. Ezdoglian, Konstantin A. Sobyanin, Vadim V. Negrebetsky, Yuri I. Baukov

**Affiliations:** Institute of Pharmacy and Medicinal Chemistry, Pirogov Russian National Research Medical University, 117997 Moscow, Russia; borozdenko@phystech.edu (D.A.B.); daria.gonchar.1999@mail.ru (D.I.G.); bogorodovavi@yandex.ru (V.I.B.); tarasdima0320@gmail.com (D.V.T.); kramarova43@mail.ru (E.P.K.); nyamomimichi@mail.ru (S.S.K.); yagolubev@gmail.com (Y.V.G.); kiseleva.67@mail.ru (N.M.K.); tatishtish@hotmail.com (T.A.S.); a.ezdoglian@gmail.com (A.A.E.); dr.konstsob@yandex.ru (K.A.S.); nmr_rsmu@yahoo.com (V.V.N.)

**Keywords:** fluoxetine, antidepressant activity, chronic unpredictable mild stress, 4-phenylpyrrolidinone-2 derivative, hormones

## Abstract

This study investigates the therapeutic potential of a new compound, potassium 2-[2-(2-oxo-4-phenylpyrrolidin-1-yl) acetamido]ethanesulfonate (Compound **I**), in depression. Willner’s chronic unpredictable mild stress model of male Wistar rats was used as a depression model. The rats were randomized into four groups, including an intact group, a Compound **I** group, a Fluoxetine group, and a control group with saline. Behavioral tests, such as the Porsolt forced swim test, hole-board test, elevated plus maze test, and light–dark box, were used to assess the animals’ conditions. Our results demonstrated that Compound **I** effectively reduced the immobilization time of rats in the forced swim test, increased orientation and exploratory behavior, and decreased the latency period of going into the dark compartment compared to the control group. Hippocampal and striatal serotonin concentrations were increased in the Compound **I** group, and the compound also reduced the level of corticosterone in the blood plasma of rats compared to the intact animals. These results suggest that Compound **I** has reliable antidepressant activity, comparable to that of the reference antidepressant Fluoxetine.

## 1. Introduction

Depression is a mental disorder characterized by a persistent decrease in mood, motor, and mental activity, as well as a reduction in or loss of the ability to feel pleasure (anhedonia). Concomitant symptoms may include sleep disturbances, decreased appetite, feelings of guilt and uselessness, development of suicidal thoughts [1]. Depressive disorders require a comprehensive therapeutic approach that includes psychotherapy, lifestyle modifications, and, in most cases, the use of antidepressants [2]. However, 30 to 60% of patients fail to respond to conventional therapy or develop pronounced adverse effects [3,4]. It should also be noted that the COVID-19 pandemic has significantly affected the epidemiology of the disease by increasing the number of patients with depression worldwide [5]. Therefore, new compounds with antidepressive properties are needed [6].

Translational studies are widely used for testing new compounds (mainly on laboratory rodents, less often on primates). There are various methods of depression modeling in rodents that differ in their difficulties and key steps of pathogenesis [7]. The most commonly used model of depression applied for potential antidepressants investigation is a method of chronic unpredictable mild stress (CUMS). The model is based on the exposure of animals to “subthreshold” stimuli, such as changes in their daily routine, crowding, a dirty cage, short-term food and/or water deprivation, etc. The stimuli change randomly, with a simulation lasting from two weeks to six weeks.

We have synthesized a new Compound **I** that has taurine as a pharmacophore; this compound, according to in silico studies, has antidepressant, nootropic, and neuroprotective effects [8]. Although taurine is a semi-essential sulfo amino acid for humans, its function in the body is very important. It is involved in membrane stabilization, osmoregulation, neuroprotection, and neuronal proliferation. We have previously shown that Compound **I** crosses the blood–brain barrier, accumulates in the cerebral cortex, and positively affects the rate of neurologic deficit elimination in an animal model of ischemic stroke [9].

The purpose of this study was to evaluate the antidepressive properties of Compound I in a CUMS model of depression and to compare its potential efficacy with the well-known antidepressant Fluoxetine.

## 2. Results

Fifty-two mature male Wistar rats participated in this study. At various stages of the study, four animals were excluded: two animals were excluded before the start of the CUMS simulation and another two animals were excluded before the start of therapy.

Before the experiment, all animals were weighed, and their psychoemotional state was assessed using the hole-board test (HBT) (RPC OpenScience Ltd., Krasnogorsk, Russia) [10]. At this stage, two animals were excluded before the initiation of depression, as they showed different activity profiles compared to the other animals (see Appendix A). Thus, 50 animals were selected for the behavioral study. These animals were distributed into two groups: those without depression stimulation and those with depression (12 animals in one group and 38 animals in another) so that the groups were similar in respect of weight and behavior activity. Then, CUMS was applied to initiate the depression. Animals were exposed to various stressful influences, such as food and water deprivation, cold stress, soft atraumatic tail clip, hot plate test, mixing of the cage animals, and light/dark cycle reversal (for a detailed description of the model, see Section 4.2).

Assessment of the development of depression in the CUMS model was conducted. To confirm the development of depression in animals, on the 14th day of modeling, the dynamics of body weight gain and a sucrose test were observed (Table 1). 

Regarding the HBT, locomotor activity sharply decreased in animals exposed to stressful stimuli (Figure 1).

During the first two weeks of exposure to stress factors, the weight gain was similar in all groups, (Figure 2), which may be because a 14-day observation is not enough for a noticeable weight loss.

These data led to the conclusion that there was an established depression. 

Thus, 2 animals were excluded from further experiments, and therapy was carried out in 36 rats. After confirming the presence of a depressive state in animals, rats with CUMS were divided into three groups for therapy (group 4—intact animals remained unchanged).
Dep_C1 (Compound_ I Dep_C1)—n = 12, therapy with Compound I 125 mg/kg (IP administration), 14 days [9,10].Dep_Flu (Fluoxetine)—n = 12, therapy with Fluoxetine 15 mg/kg (oral administration).Dep_NaCl (control group treated with saline)—n = 12, IP administration of physiological saline.Int (intact animals)—n = 12, not exposed to depressive stimuli and not receiving any medications. 

The therapy began during the stress exposure. This better simulates real clinical practice, where patients begin to take antidepressants while still under stress. We chose a 14-day administration because it has been shown that this administration regimen is used for fluoxetine, a comparator drug, and gives the necessary pharmacological response in animal tests [11,12,13].

### 2.1. Behavioral Tests

We investigated the effects of Compound I on the development and progression of depression in the model of CUMS in rats. The study design is shown in (Table 2).

Weight loss or food anhedonia is one of the most striking symptoms of the development of a depressive state. According to the literature data, in the CUMS model, under the influence of stress factors in rats, body weight significantly decreases. In our study, there were no statistically significant differences in body weight gain in animals by the beginning of the third week (Figure 2). However, starting from the third week, there was a significant increase in weight gain in intact animals (Figure 2).

Starting from the third week, as a result of further exposure to depressive stimuli, weight gain decreased. After the start of therapy, differences began to be observed between the group Dep_NaCl (physiological saline) and the intact group. After 2 weeks of treatment, this difference was observed between the group of intact animals and all groups of animals.

In the sucrose test, already a week after the start of therapy, sweet water consumption increased by 8.8% and by 14% in Dep_C1 and Dep_Flu, respectively, relative to sucrose consumption before therapy. For animals of Dep_C1 and Dep_Flu, the increase in consumption of sweet water continued throughout the entire therapy, although it did not reach the level of that of intact animals (Table 3). At the same time, Dep_NaCl animal consumption of sweet water throughout the entire therapy remained at a level of depression that was 23% lower than that of intact animals. There was no statistically significant difference between intact and NaCl-treated animals; however, there was a trend (Table 3).

Further, to evaluate the effectiveness of antidepressant therapy, the following tests were performed: the Porsolt test, the light–dark box test (LDB), and the elevated plus maze test (EMT).

The Porsolt test is the main test for determining the depressive phenotype of behavior, aimed at assessing the time of immobility (immobilization) when the animal does not make active attempts to get out—jumping or diving to leave the installation [14]. This test has the highest predictive validity since the antidepressant use reduces the time of immobility, that is, it eliminates the feeling of hopelessness and inability to get out of the cylinder [14].

Generally, depressed rats show apathy and do not try to get out of the water. The time of immobility is the criterion for the magnitude of the antidepressive effects of the compounds. In this study, the test showed a significant reduction in the immobilization time in the Compound **I** and Fluoxetine groups compared to the control groups (Figure 3).

In the Porsolt test, the time of immobilization of animals in group Dep_NaCl increased, which indicates the presence of a depressive state. We observed similar behavior in the LDB and EMT tests (Figure 4).

The light–dark box test (LDB) and the elevated plus maze test (EMT) are based on the fact that animals prefer to stay in a dark place. More severe depression is associated with the prolongation of the latency of entrance to the dark compartment and the increased time spent in the light compartment. In the Compound **I** and Fluoxetine groups, the entry latency and residence time in the light compartment were significantly lower than in the control group and did not differ from intact animals. Rats under the influence of the test substance and the reference antidepressant, in the presented tests, restored the normal reflex act of fear of bright spaces and also showed an increased exploratory activity (Figure 4).

The hole-board test (HBT) is used in experimental pharmacology to assess orientational and motor activity. The vertical activity in the experimental group Dep_C1 was significantly higher than in the Dep_NaCl group. The number of crossed sectors in the group Dep_C1 was comparable to that in the group Del_Flu and significantly higher than that in the Dep_NaCl (Figure 5).

On Day 31 of the study, the OFT test was performed to assess orientation and motor activity. The results had a similar trend to the HBT test (see Appendix A for details).

### 2.2. Monoamine Concentration Studies

According to the monoamine-related hypothesis, depression is associated with a reduced content of serotonin and an impaired ratio of other monoamines, particularly dopamine; therefore, the levels of these substances were evaluated in the hippocampus and striatum of rats. The hippocampal concentration of monoamines was assessed as the volume of the hippocampus decreases in depressed patients [15]. Impaired neurogenesis and synaptic dysfunction in the hippocamp may play a key role in the pathophysiology of depressive disorders. In turn, the striatum was chosen for the analysis of monoamines as the part of the brain responsible for motor functions in animals and in depressive disorders is also actively involved in the pathological process [16]. Compound I therapy led to a significant elevation of serotonin in both the striatum and the hippocampus, and the striatal dopamine level tended to increase. The data obtained in the group Dep_C1 were comparable with those in the Dep_Flu group (Figure 6).

During physical or emotional stress, the hypothalamus–pituitary–adrenal (HPA) axis is activated. The HPA axis is a complex system that is activated during physical or emotional stress. This process involves the secretion of two key hormones by the hypothalamus, corticotropin-releasing hormone (CRH) and arginine vasopressin (AVP), which then act on the pituitary gland to increase the release of the adrenocorticotropic hormone (ACTH). In turn, the adrenal cortex is stimulated by ACTH to produce corticosterone, the primary stress hormone.

Plasma corticosterone levels were measured after completion of the entire experiment (post-mortem). In the CUMS model, rats were exposed to stress which resulted in elevated plasma levels of corticosterone in saline-treated animals ((Dep_NaCl) (Figure 7a)). However, Compound **I** (Dep_C1) was found to effectively reduce the concentration of corticosterone compared to that in both intact (Int) and physiological saline animals (Dep_NaCl) (Figure 7a). Interestingly, there were no significant differences in adrenocorticotropic and corticotropin-releasing hormone concentrations between all groups (Figure 7b,c).

## 3. Discussion

Depressive disorder is a complex and multifactorial illness with an uncertain pathogenesis. One theory suggests that it is caused by abnormal levels of monoamines (such as serotonin, dopamine, norepinephrine, and their precursors) in various parts of the brain [17]. This hypothesis has been the basis for developing many of the current antidepressant drugs. However, even with treatment progress, current antidepressants do not always alleviate all depression symptoms, such as sleep disorders, anxiety, and fatigue [18]. The involvement of the hypothalamus–pituitary axis has also been shown, as patients with depressive disorders often have elevated cortisol levels [19]. Moreover, there is evidence that the activation of peripheral immune response and neuroinflammation can contribute to the development of depressive disorders [20]. Based on the available data, it can be assumed that depression pathogenesis involves disruptions in the interactions between the nervous, endocrine, and immune systems and that a comprehensive therapeutic approach is required, which has been clinically proven. For instance, a depressive disorder may develop after a viral infection [21], malignancies [22], or diabetes mellitus [23], even in patients without a family history of depression.

There is evidence that compounds based on the pyrrolidine ring have an anticonvulsant effect with anxiolytic and antidepressant properties [24]. The biological activity of compounds based on the pyrrolidin-2-one heterocycle is highly dependent on conformational changes in the chiral center and on the introduced pharmacophores [25,26,27]. To enhance the potential therapeutic properties of pyrrolidin-2-one, we proposed the introduction of a taurine residue, which has neuroprotective properties and can improve exploratory activity. Several studies have suggested that taurine may have antidepressant and anxiolytic effects. For example, in rats with ethanol-induced CNS depression, taurine was found to increase sleep time, and in a CUMS model, taurine administration reduced anxiety and restored sucrose intake [28,29]. Using a “silyl” method for N-alkylation of lactams, we synthesized Compound **I**, which incorporates a taurine residue [9,30].

The objective of this study was to determine whether taurine modification of the pyrrolidin-2-one heterocycle (Compound **I**) has antidepressant activity and compare its effects to the widely used antidepressant Fluoxetine. We selected stimuli to model depression using the chronic unpredictable mild stress method, which proved effective in inducing depressive behavior by the second week of the experiment. In our CUMS model, 95% of the animals developed depression. This was confirmed by the HBT test, where rats exhibited reduced motor activity, and the LDB, which showed increased anxiety levels (as evidenced by a longer latency to enter the dark compartment). Additionally, there was a trend towards weight loss compared to the intact group after 21 days of modeling. These tests proved effective in identifying depressive behavior in most of the rats, except for two particularly active rats that did not exhibit any signs of depression. 

The primary behavioral test demonstrating the antidepressant properties of the compounds was the forced swimming test. In our study, rats treated with Compound **I** and Fluoxetine significantly reduced their immobility time and did not stop trying to escape until the end of the test. Based on the results of the OFT, HBD, LBD, and EMT behavioral tests, the tested compound exhibited antidepressant activity by increasing motor reactions and exploratory behavior while decreasing apathy and anxiety. The antidepressant effect was comparable to that of the selective serotonin reuptake inhibitor Fluoxetine (Figure 3, Figure 4, Figure 5 and Figure 6).

When evaluating the behavioral responses of animals treated with Compound **I** and Fluoxetine, they exhibited greater motor activity and exploratory behavior compared to the control groups (Figure 5). These results suggest that Compound **I** can reduce depression symptoms such as apathy and hypokinesia. In the “light–dark box” and “elevated plus maze” tests, depressed rats lost their preference for the dark side due to apathy, indicating a lack of survival reflexes. However, animals treated with Compound **I** and Fluoxetine regained their ability to hide, as demonstrated by a decrease in the latency period for entering the dark compartment and reduced time spent in the lighted arm of the maze during the EPM test. In contrast, animals treated with the saline solution remained apathetic and stayed in the lighted compartments.

Postmortem data from animal and human studies have shown that dysfunction in the monoaminergic neurotransmitter systems, such as serotonin (5-HT), norepinephrine (NE), and dopamine (DA), plays a key role in the development of depression [31]. However, conflicting results exist regarding the effects of chronic stress on the levels of these neurotransmitters. Our study supports the findings that chronic stress leads to a decrease in 5-HT levels in the hippocampus [32,33].

Our behavioral test results showed that treatment with a serotonin reuptake inhibitor (SSRI) fluoxetine had a positive effect in this model, indicating that the balance of serotonin plays a crucial role in the development of depression in CUMS. We also demonstrated that administration of Compound **I** resulted in the restoration of serotonin levels in both the striatum and the hippocampus in rats, which was comparable to the levels observed in rats treated with Fluoxetine. This suggests that the two drugs may share similar mechanisms of action. In future studies, we plan to investigate the levels of homovanillic acid and the precursors of dopamine and serotonin. For a long time, the activation of the hypothalamic–pituitary system was considered the key step in the pathogenesis of depressive behavior in the CUMS model rather than neuroinflammatory changes [34]. In our study, we demonstrated an increase in corticosterone levels in the blood plasma of animals subjected to CUMS without therapy, which was significantly reduced upon treatment with Compound **I** and Fluoxetine. The ability to influence depressive behavior by normalizing the balance of monoamines and reducing stress hormone levels has been confirmed by some studies on Fluoxetine [35], but not all researchers agree as data suggest that Fluoxetine may increase corticosterone levels in some cases [36,37]. Our study provides direct evidence of a link between the reduction of depressive behavior, the increase in monoamine concentrations, and the decrease in corticosterone levels upon treatment with Compound **I** and Fluoxetine.

## 4. Materials and Methods

### 4.1. Chemistry

Compound **I** was synthesized according to the previously proposed method [10]. 

This method involves a four-step approach starting with the alkylation of lactam **1** with ethyl chloroacetate in the presence of sodium hydride (compound **2**), followed by alkaline hydrolysis (compound **3**), acid esterification (compound **4**), and treatment with taurine to obtain Compound **I** (Figure 1). This method is suggested for industrial production due to its higher yields and less contamination.

The yields of the products were 80–90%.

### 4.2. Animals Models

The behavioral study was conducted on 52 mature male Wistar rats with an average weight of 220 ± 12 g, obtained from the animal nursery of FSBI Federal Research Center for Cytology and Genetics of the Siberian Branch of the Russian Academy of Sciences in Novosibirsk. Before modeling, 2 animals were excluded based on weight and activity (see Appendix A). The animals were housed in a standard facility at N.I. Pirogov Russian National Research Medical University with automatic day and night cycles (08:00–20:00—“day”, 20:00–08:00—“night”) and optimal temperature (20–24 °C) and humidity levels (45–65%), ensuring their comfort and meeting the standards of Directive 2010/63/EU on the protection of animals used for scientific purposes. The experiments were approved by the Commission for the Care and Use of Animals at Pirogov Russian National Research Medical University (Protocol No. 08/2021).

The depression modeling of CUMS is presented in Table 4.

The group of intact animals was not exposed to stressful conditions. To confirm the depressive state, a sucrose test was performed, and on Day 14, the hole-board test was repeated. Based on the test results, 2 animals were excluded.

### 4.3. Treatment (Administration of Compounds)

All compounds were administered for 14 days. The dose of Compound **I** (125 mg/kg) was chosen based on the results of our previous studies—pharmacokinetic studies—and the studies of the beneficial effects of the substance on neurologic deficits [9,38]. The intraperitoneal route of administration was chosen as an equivalent of IV infusion (for longer administration). Fluoxetine was selected as the comparator substance because selective serotonin reuptake inhibitors are currently used as first-line drugs for the treatment of depression and are widely applied in in vivo studies. The therapeutic dose for rats with short-term treatment was 15 mg/kg [10].

### 4.4. Behavioral Test for Efficacy Measurement

Anhedonia, a significant reduction or loss of the ability to experience pleasure, is a hallmark symptom of depressive disorder. In rat studies, the sucrose test is the most commonly utilized method to evaluate the severity of anhedonia [38]. To conduct the test, animals are subjected to water (6 h) and food deprivation (12 h) and then placed into cages (2 rats per cage). The test is performed from 9:00 to 21:00, during which time the animals are only allowed access to drinking water or a 2% sucrose solution for 12 h. The bottles are weighed before and immediately after the test. 

Additionally, in assessing depression using the behavioral hole-board test, decreased vertical activity (rearing and hiding episodes—not more than 15) and horizontal activity (not more than 20 crossed sectors) are used as criteria.

### 4.5. Behavioral Tests

The following tests were used for the assessment of general motor activity orientation and exploratory behavior (see Table 2).

The hole-board test (HBT) (RPC OpenScience Ltd., Krasnogorsk, Russia) was conducted using a modified technique [39] before the study to create groups with identical behavior. The number of crossed sectors, vertical activity, speed, and distance traveled were monitored on Day 14 of the experiment to confirm the development of a depressive state and on Day 30 of the experiment to assess the psychoemotional state. The open-field test (RPC OpenScience Ltd., Krasnogorsk, Russia, Russia) was carried out on Day 31 of the experiment using the method described in [40] for 5 min. The parameters tested were the same as in the HBT. Two similar tests were used to exclude the memorization and habituation of the animals to the settings. The light–dark box test (LDB) (Neurobotics, Zelenograd, Russia) was conducted as described earlier [41]: the latency of entrance to the dark compartment, the time spent in the light compartment, the number of instances of peeking out from the dark compartment, the number of rearing episodes were recorded for 3 min. The test was performed on Day 14 of depression modeling to assess the development of depression and on Day 33 of the experiment. The elevated plus maze test (EMT) (RPC OpenScience Ltd., Krasnogorsk, Russia) was conducted using previously described methods [42]. The time spent and the number of visits to the light compartment, the number of rearing episodes, hanging, peeking out, and the number and time of grooming and fading episodes were recorded on Day 34 of the experiment. The Porsolt forced swim test (FST) was conducted for 5 min. The time and the number of episodes of active swimming and immobility and the number of jumps and dives according to [43] were recorded. The test was carried out on the last day of the experiment.

### 4.6. Monoamine Concentration Studies (LC-MS)

To evaluate serotonin and dopamine concentrations in the brain structures, the rats were decapitated with a guillotine and the hippocampus and striatum were isolated on ice. The samples were studied on a high-performance liquid chromatography apparatus (Shimadzu LC-20AD, Shimadzu, Kyoto, Japan). The samples were stored at −80 °C. To carry out the analysis, the weighed samples were placed in 2 mL test tubes with seal caps. The weights of the samples were 35 ± 10 mg for the striatum and 41 ± 14 mg for the hippocampus. Caffeic acid was used as an internal standard.

A total of 240 mL of deionized water, 30 mL of caffeic acid solution (500 ng/mL), and 30 mL of formic acid were added to the test samples. The resulting mixture was homogenized on a Precellys Evolution apparatus (Bertin Technologies, Montigny-le-Bretonneux, France) at 5000 rpm for 1.5 min. An amount of 300 µL of chloroform was added to the resulting solution and mixed on Multi-Vortex Biosan (Riga, Latvia) for 30 s, followed by centrifugation (CM-50; ELMI; Riga, Latvia) for 5 min at 12,000× *g*. An amount of 300 µL of chloroform was added to the resulting supernatant and mixed on a Multi-Vortex for 30 s, followed by centrifugation (CM-50; ELMI; Riga, Latvia) for 3 min at 12,000× *g*. The supernatant was transferred to 300 µL vials for subsequent testing.

Two calibration curves were used for the determination of monoamine concentrations of the hippocampal and striatal samples. The brain tissue samples used to construct the calibration curve were kept at ambient temperature for several hours to ensure complete degradation of dopamine and serotonin. To construct calibration curves, 210 mL of deionized water, 30 mL of caffeic acid solution (500 ng/mL), 30 µL of a mixture of serotonin and dopamine reference standards, and 30 µL of formic acid were added to the suspension. Further sample preparation was identical to that described above.

Supelco Ascentis (Sigma-Aldrich; St. Louis, MO, USA) C18 25 cm 4.6 mm column was obtained. Mobile phase A: 0.2% formic acid in deionized water; mobile phase B: 0.2% formic acid solution in methanol; 0–0.5 min, A—95%, B—5%; 0.5–7 min linear gradient to A—20%, B—80%; 7.01–10 min, B—100%; 10.01–17 min, A—95%, B—5%. The volume of the pumped mobile phase was 0.8 mL/min. The temperature of the sample cell was 9 °C. The temperature of the thermostat was 40 °C. The injection volume was 20 mL.

The compounds were analyzed using the electrospray ionization (ESI) method. Positive ionization was used for testing. The interface voltage was 3.5 kV. The temperature of the heating unit was 400 °C; the desolvation temperature was 250 °C; the spray gas flow rate was 3 L/min; the drying gas flow rate was 15 L/min; and the gas CID pressure was 60 kPa. The MRM transitions used were 154.05→137.10 for dopamine, 177.10→160.10 for serotonin, and 178.95→134.95 for caffeic acid.

Sample collection. On the last (36th) day of the study, the animals were humanely euthanized by guillotine. Blood, hippocampus, and striatum samples were collected. Blood was separated after centrifuging at 3000 rpm for 10 min at 4 °C and was kept at −20 °C. The hippocampus and striatum were extracted on ice and frozen in liquid nitrogen.

Calibration curves and typical chromatograms are presented in Appendix A for details.

### 4.7. Measurement of Hormones and Inflammatory Factors

Enzyme-linked immunosorbent assay (ELISA) was used to measure serum hormone levels. Serum concentrations of corticosterone (Cort), corticotropin-releasing hormone (CRH), and adrenocorticotropic hormone (ACTH) (#ELK8633, #ELK 2606, #ELK2414, ELK Biotechnology Co., Ltd., Wuhan, China) were measured using commercial ELISA kits. Serum values were determined without any dilution. All experimental steps were performed according to the manufacturer’s instructions. The plates were precoated with Cort, CRH, and antibodies to PRL, respectively. Optical density values were recorded at 450 nm, and concentrations were calculated from the standard curve.

Statistical analysis was performed using STATICTICA software version 12.0. The Kolmogorov–Smirnov test was used to assess the assumption of normal behavioral and biochemical findings. All values are presented as mean ± standard error of the mean (SEM), and *p*-values less than 0.05 were considered significant. One-way analysis of variance was used to analyze all behavioral and biochemical data, followed by post hoc analysis using the Fisher test for LSD.

## Data Availability

Data are contained within the article and Appendix A.

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
