# Peer review of "The Antidepressant Activity of a Taurine-Containing Derivative of 4-Phenylpyrrolidone-2 in a Model of Chronic Unpredictable Mild Stress"

_ijms, 2023, doi:10.3390/ijms242316564_

Round 1

Reviewer 1 Report (New Reviewer)

Comments and Suggestions for Authors

Comments:

Abstract:

Willner's                                                                                                                                   12

chronic unpredictable mild stress model of depression in 48 male Wistar rats was used a depression           13

model.” – modify sentence to avoid repetition of depression e.g., chronic unpredictable mild stress in 48 male Wistar rats was used to modulate depressive-like behaviour.

“including an intact group…… control group receiving saline” --- better control group treated with saline……CUMS treated receiving saline

Introduction:

“The method is based on the expo-                                                                                               42

sure of animals to “subthreshold" stimuli, such as changes in the daily routi….”  --- replace “method” to “model”

“”

Results:

In abstract you wrote “Willner's 12

chronic unpredictable mild stress model of depression in 48 male Wistar rats was used a depression 13

model.” In results “The behavioral study was conducted on 52 mature male Wistar rats. Of which, 4 rats 57

were excluded before the start of therapy, as they showed deviant behavior before ran- 58

domisation” ---- my question is in the end how many rats you used in CUMS 52 and just 48 were carried out of behaviour or 52 but statistical analysis includes 48.

start of therapy,” --- in this case you are talking about CUMS or drugs treatment???

“then CNLS was” ---you mean CUMS??

Assessment of the development of depression in the CNLS modelTo confirm the de-                                     69

velopment of depression in animals, on the 14th day of modeling, the dynamics of body” --- CNLS model ???? what it is???

So, weight loss and a reduced sucrose consumption are in- 73

dicators of the development of anhedonia (i.e., a decrease or loss of the ability to experi- 74

ence pleasure), and a decrease in locomotor activity indicates a reduced interest and curi- 75

osity. After two weeks, 2 rats from the CNLS group were excluded from the study, as they 76

didn’t show any signs of depression.” --- Rewrite sentence…. Do not start from “So”…..ones more used “CNLS”

Table 1.” --- can be improved. It will be more readable when you’ll start from CNT, CUMS responders and CUMS non-responders etc.

Figure 1” ----- there is no “*” on the graph….

After confirming the presence of a depressive state in animals, rats with CNLS were 87

divided into 3 groups for therapy (group 3 - intact animals remained unchanged). 88

Dep_C1 (Compound_1 Dep_C1) – n=12, therapy with Compound I 125 mg/kg 89

(IP administration). 14 days; 90

Dep_Flu (Fluoxetine) – n=12, therapy with Fluoxetine 15 mg/kg (oral admin- 91

istration). 92

Int (intact animals) – n=12, not exposed to depressive stimuli and not receiving 93

any medications 94

Dep_NaCl (physiological saline) – n=12, IP administration of physiological sa- 95

line. 96

The therapy began durin” ----- are you sure that FLX was administrated orally and C1 i.p. in those dosages???? It is not a topsy-curvy?? Why you not treated intact rats with saline ??? you wrote that “We 98

chose a 14-day administration because it has been shown that this administration regimen 99

is used for fluoxetine,” is it means that all treatment was in 14 days???

The first paragraph of “Results” is very confusing, rewrite that please.

Table 2. Study Design” is not readable. Is it means that CUMS was only for 14 days???  In materials and methods you have 21d of stress exposure

“Figure 2.” ---- use correct nomenclature for units ---- [g]

sumption of sweet water continued throughout the entire therapy, although it did not 119

reach the level of intact animals (Table 3.). At the same time, Dep_NaCl animalconsump- 120

tion of sweet water throughout the entire therapy remained at a level of depression that 121

was 23% lower than that of intact animals (Table 3” --- sweet water is better to sucrose consumption, space in “Dep_NaCl animalconsump”

Table 3.” --- there is no statistical significances between groups C1; FLU vs Dep_NaCl?????

“s not make active attempts to get out - jumping or diving to leave the installation) [12]. 130

[Dawson C.A., Horvath S.A. Swimming in small laboratory animals // Med. Sci. Sports. – 131

1970. Vol. 2 (2). – P. 51–78]. This” ------ remove surplus citation

“to get out of the installation [12].” --- replace word “installation” with different one…..

Figure 3.” ---- graph is to overstretched…. Normalize name of groups in all of tables and graphs… in fig 3 you used Compound 1 …. NaCl 0.9% instead of Dep_C1….Int in table 3…. Is NaCl group not significant to Intact???

“Since depressed rats show apathy and do not attempt to get out and freeze on the 135

water's surface.” ---- rewrite sentence

Figure 4. ” ---- nomenclature of units --- [s]; normalize groups name; add statistical test you used

Figure 5.” normalize groups name, in text you used “group Dep_C1 153

was significantly higher than in the Dep_NaCl.”

“2.2. Monoamine concentration studies 157

On day 31 of the study, the OFT test was performed to assess orientation and motor 158

activity. The results had a similar trend to the HBT Test (see SM Fig.S1 for details).” --- you’re writing about OFT in “2.2. Monoamine concentration studies”, why???

“Figure 6.”----- units in brackets…. Don’t repeat HPLC every time, just mention it on ce in description e.g. Dopamine and serotonin concentration in striatum and HP using HPLC; a)……

Why you used in treated experiment one-way ANOVA not two-way….

Discussion:

In our CUMS model 95% 219

of the animals developed depression.This was confirmed by the HBT test, where rats ex- 220

hibited reduced motor activity, and the LDB which showed increased anxiety levels (as 221

evidenced by a longer latency to enter the dark compartment). Additionally, there was a 222

trend towards weight loss compared to the intact group after 21 days of modeling. To 223

assess the depressive state of the animals, we used simple and quick behavioral tests such as the sucrose preference test, the dark-light box, and the hole-board test.” ---- you’re repeated twice the same thoughts

In our study, rats treated with Compound I and 229

Fluoxetine significantly reduced their immobility time and did not stop trying to escape 230

until the end of the test. Based on the results of several behavioral tests, the tested com- 231

pound exhibited antidepressant activity by increasing motor reactions and exploratory 232

behavior while decreasing apathy and anxiety. The antidepressant effect was comparable 233

to that of the selective serotonin reuptake inhibitor Fluoxetine” ---- write it better…. Several beh test--- name them

These results suggest that Compound I can mitigate depression 237

symptoms such as apathy and hypokinesia. I” ---- so it is antidepressant or caused depressive-like behaviour

Material and methods:

“treatment ” is not clear explained. Is C1 was infused/injected for 5 days or 14 days???

SPT ---- how you recognized drinking volume of sucrose/water if rats were 2 in cage during the SPT protocol???

Comments on the Quality of English Language

 Moderate editing of English language required

Author Response

Dear reviewer! This is our first experience of presenting an article of this kind in your journal. We are grateful to you for his meticulous work and comments, which related to improving the presentation of not only factual material, but also literary corrections.
We hope that you will be satisfied with the changes made to the article, and our answers

Reviewer 2 Report (New Reviewer)

Comments and Suggestions for Authors

Authors should ensure the correct nomenclature for the analytical technique is used. "HPLC" in the subheading may need to be revised to "LC-MS" for clarity and accuracy.
Authors are requested to include Multiple Reaction Monitoring (MRM) and Total Ion Chromatogram (TIC) data for each sample as supplementary information to support the results.
 Clarification is needed regarding the consistent grouping of subjects as '12' and '38.' Table 4 should specify the stress factors, not the numerical grouping. Authors are encouraged to provide justification for the numbering within the groups.

The toxicology test for "Potassium 2-[2-(2-oxo-4-phenylpyrrolidin-1-yl) acetamido]ethanesulfonate (Compound I)" should be included in the manuscript.
Authors are requested to include the synthetic route for the Compound I in the manuscript, specifically within the materials and methods section, a scheme is mentioned but only a structure is presented. 

Comments on the Quality of English Language

Minor editing of English language required

Round 2

Reviewer 1 Report (New Reviewer)

Comments and Suggestions for Authors

Dear authors;

Thank you for your answers to my comments. I accept almost all explanations, one needs to be approved. Can you add reference to "Dep_C1 (Compound_ I Dep_C1) – n=12, therapy with Compound I 125 mg/kg (IP administration). 14

days;

Dep_Flu (Fluoxetine) – n=12, therapy with Fluoxetine 15 mg/kg (oral administration).

Dep_NaCl (control group treated with saline) – n=12, IP administration of physiological saline.

Int (intact animals) – n=12, not exposed to depressive stimuli and not receiving any medications." dosages. I mean that please, cite papers for Dep_C1 and Dep_Flu where you show that the way of administration and amount of drugs are working in presented paradigms.  Comments on the Quality of English Language

Still needs some correction, especially with typos.

Author Response

We are grateful to the reviewer for his meticulous work and comments, which related to improving the presentation of not only factual material, but also literary corrections. 

Reviewer 2 Report (New Reviewer)

Comments and Suggestions for Authors

Please accept in the present form. 

Author Response

We are grateful to the reviewer for his meticulous work and comments, which related to improving the presentation of factual material.

This manuscript is a resubmission of an earlier submission. The following is a list of the peer review reports and author responses from that submission.

Round 1

Reviewer 1 Report

Comments and Suggestions for Authors

The paper by Borozdenko and co-authors devoted to study of antidepressant properties of new taurine containing derivative of 4-phenylpyrrolidone. The findings are potentially interesting but the study has the methodical flaws and the manuscript is written carelessly. So, I have serious concerns that led to my negative decision regarding the manuscript, as follows:

Major:

1) CUMS interventions were started before the start of drug administration. Moreover, stressful interventions continued after the administration of drugs was started. I don't think this design is good. With the chosen approach, it is impossible to say with certainty whether the administration of compound 1 prevents the development of depressive-like behavior or eliminates its consequences. So, authors should clearly justify their chosen design.

2) The adequate description of statistical data (with exception of legends for figs 4 and 5) is absent throughout the “results” section. The description of statistical methods is also absent in the “material and methods” section. So, the important information regarding choice of statistical methods based on normality of distribution, degrees of freedom, outliers, etc., is unknown for readers. For example, did the Bonferroni correction performed under analysis of Hole-board test results since this test was performed triple on the same animals?

3) The data on the body weight should be re-analyzed with ANOVA test for repeated measures and presented as linear diagrams. The changes in body weight measured in the same animals at different time points are a dynamic parameter and the analysis of these data as independent variables is inappropriate.

4) The results of the Open-field test are absent in the text while this test is mentioned in the “Materials and methods”.

5) The authors should provide (at least in the appendix) the data of HBT conducted on the zero and 14th days of the experiment. This would help to understand whether there were initially differences between groups when performing the test.

6) Why did the authors not provide data on immobility and grooming time for HBT, while this was parameters stated in the materials and methods (line 247)?

7) The figure 2 duplicates the figure 1. Well, it is about carelessness.

8) It is not clear, why authors used both light-dark box test and elevated plus maze test. In general measures these tests duplicate each other. The description of results from LDB and EMT also is not clear. Parameters of which test do the authors mean (lines 91-92)?

9) The authors interpret the animals' preference for the light compartment in the LDB as a sign of apathy (lines 172-173). Conversely, the animals' preference for the dark compartment after drug exposure is interpreted as an anxiolytic effect (lines 184-186). The LDB is based on an approach-avoidance conflict between exploration of novel environments and avoidance of brightly lit, open spaces. Considering that the number of studies indicates the anxiolytic effect of SSRIs in rats as preference of open bright compartment, the authors should discuss their observation more detailed.

10) Was it really only one animal that showed no signs of depressive-like behavior (line

177)? Were any other animals excluded from next testing?

11) Why were the hippocampus and striatum chosen to measure monoamine concentrations? Why only these brain structures, and not, for example, the midbrain? Authors should justify the choice of brain structures for analysis.

12) Graphs in Figure 4 are difficult to read. I recommend splitting Figure 4(a) into two - one for the 5-HT level, one for the dopamine level. In addition, the p values indicating differences between the groups of compounds and intact group or the NaCl group should be indicated by different signs, for example (*) and (#).

Minor:

1) The legends for all figures should be more informative.

2) Table 5 provides more valuable information than the Scheme 1. I think the Scheme 1 should be deleted.

3) The Figure 5 need to be reorganized. Now two parts of the picture are located on different pages - this is unacceptable

Reviewer 2 Report

Comments and Suggestions for Authors

This paper examines the moderately stress-induced depressive effects of a novel taurine derivative.

Comment 1: The author is examining the depressive effects induced by various stresses and administering drugs from the onset of the stress load. This schedule should also be noted in the figure.

Comment 2: The statistical treatment is inappropriate. The LSD is being compared between multiple groups, but the LSD is not appropriate. Other statistical treatments should be implemented.

Comment 3: The same figure is included. It must be carefully checked before submission. You are submitting something incomplete to the reviewers